# Experimental Study and Modeling of Beer Dealcoholization via Reverse Osmosis

**DOI:** 10.3390/membranes13030329

**Published:** 2023-03-13

**Authors:** Áron Varga, Eszter Bihari-Lucena, Márta Ladányi, Beatrix Szabó-Nótin, Ildikó Galambos, András Koris

**Affiliations:** 1Department of Research and Development, Pécsi Brewery, Alkotmány utca 94., H-7624 Pécs, Hungary; 2Department of Food Process Engineering, Hungarian University of Agriculture and Life Sciences, Ménesi út 44., H-1118 Budapest, Hungary; 3Department of Agricultural Business and Economics, Hungarian University of Agriculture and Life Sciences, Villányi út 29-43., H-1118 Budapest, Hungary; 4Department of Bioengineering and Fermentation Technology, Hungarian University of Agriculture and Life Sciences, Ménesi út 45., H-1118 Budapest, Hungary; 5ICON PLC, Szépvölgyi út 39., H-1037 Budapest, Hungary; 6Department of Applied Statistics, Hungarian University of Agriculture and Life Sciences, Villányi út 29-43., H-1118 Budapest, Hungary; 7Department of Fruit and Vegetable Processing Technology, Hungarian University of Agriculture and Life Sciences, Villányi út 29-43., H-1118 Budapest, Hungary; 8Department of Soós Ernő Research and Development Center, University of Pannonia, Zrínyi Miklós utca 18., H-8800 Nagykanizsa, Hungary

**Keywords:** alcohol-free beer, dealcoholization, low-alcohol beer, membrane separation processes, modeling, reverse osmosis

## Abstract

The goals of the present investigation are to study and to model pale lager beer dealcoholization via reverse osmosis (RO). Samples were dealcoholized at a temperature of 15 ± 1 °C. An Alfa Laval RO99 membrane with a 0.05 m^2^ surface was used. The flux values were measured during the separations. The ethanol content, extract content, bitterness, color, pH, turbidity, and dynamic viscosity of beer and permeate samples were measured. The initial flux values were determined using linear regression. The initial ethanol flux ($J_{EtOH\;0}$) values were calculated from the initial flux values and the ethanol content values. A 2^P^ full factorial experimental design was applied, and the factors were as follows: transmembrane pressure (TMP): 10, 20, 30 bar; retentate flow rate (Q): 120, 180, 240 L/h; $J_{EtOH\;0}$ was considered as the response. The effect sizes of the significant parameters were calculated. The global maximum of the objective function was found using a self-developed Grid Search code. The changes in the analytical parameters were appropriate. The TMP had a significant effect, while the Q had no significant effect on the $J_{EtOH\;0}$. The effect size of the TMP was 1.20. The optimal value of the factor amounted to TMP = 30 bar. The predicted $J_{EtOH\;0}$ under the above conditions was 121.965 g/m^2^ h.

## 1. Introduction

The scope of this research was to study and model beer dealcoholization (BDA) via reverse osmosis (RO).

Beer is one of the most popular beverages all over the world [1]. Moderate beer consumption has health benefits, but these benefits are restricted by the negative consequences of the ethanol (alcohol) content of beer. However, there is the potential to reduce the ethanol content of beer through innovation [2]. The production of beers with a reduced ethanol content is a fast-growing segment in the global beer market [2].

The legal definitions of low-alcohol beer (LAB) and alcohol-free beer (AFB) vary from country to country [3]. For example, in Hungary (the country of this study), the ethanol content of LAB must be between 0.51 and 1.50% (*v*/*v*), and the AFB must contain a maximum ethanol level of 0.50% (*v*/*v*) [4].

There can be several reasons for LAB or AFB production. The reasons are as follows: an increase in the overall production by introducing new products in countries with highly competitive markets, providing beer consumers with products prior to or during their activities (driving motor vehicles, operating machinery, or doing sports) or under conditions (pregnancy or medication) irreconcilable with alcohol consumption, penetrating the beverage markets in countries where alcohol consumption is forbidden for religious reasons [5].

The aim of LAB or AFB production is to reduce the ethanol content of beer while maintaining other characteristics [2]. There are different methods for LAB or AFB production. Figure 1 shows the schemes of LAB and AFB production methods (based on [3,5,6,7,8]).

As can be seen in Figure 1, one of the groups of methods is the membrane separation process.

The membrane separation processes provides promising alternatives for separating the ethanol after the fermentation process, and includes advantages such as a lower energy consumption, no chemical additives, and operation at mild temperatures, thereby reducing the impact of heat on the product [9]. In this study, a membrane separation process for BDA was investigated, namely RO.

However, it is very important to compare the membrane separation processes, especially RO, with the different low-alcohol beer (LAB) and alcohol-free beer (AFB) production methods. Table 1 shows a comparison of the installation cost, operating cost, and quality of the final beer using the low-alcohol beer (LAB) and alcohol-free beer (AFB) production methods (based on [10]).

As can be seen in Table 1, BDA via RO is a slightly costly method, but the quality of the final beer is good compared to the other methods. Due to the higher costs, this method is a feasible solution for larger breweries with capital.

It is important to note that the scopes of several studies include BDA via RO [11,12,13,14]. However, none of the scopes of these studies have included physical and mathematical modeling, or the mathematical optimization of BDA via RO. Thus, our study, with its scope, fills these research gaps. This was the reason for why our research was carried out. Furthermore, the Alfa Laval RO99 membrane has not been used for BDA according to the recent literature, and the values and the ranges of the operating parameters in our study were different from the values and ranges that can be found in the recent literature. In addition, our beer sample (Soproni Klasszikus pale lager beer (HEINEKEN Hungária, Sopron, Hungary)) was different from the beer samples that were dealcoholized in the recent literature [11,12,13,14].

The most important parameters of the BDA via RO are the permeate flux and the ethanol concentration in the permeate. These parameters can be combined into one parameter: ethanol flux [15,16,17].

The operating parameters affect the permeate flux and the ethanol concentration in the permeate [11]; therefore, the operating parameters also affect the ethanol flux. Thus, the optimization of the operating parameters is essential to achieve ethanol flux enhancement. A full factorial experimental design can be used successfully to optimize the operating parameters of membrane separation and to study the process [18,19,20,21] with a minimal number of experiments [22].

The goals of the present investigation were (i) to determine the analytical parameters of beer and permeate samples (ethanol content values for the physical modeling); (ii) to determine the hydrodynamic parameters of the membrane separations for the physical modeling; (iii) to calculate the ethanol flux values of the membrane separations for the response (physical modeling) of the experimental design; (iv) to analyze the experimental design (mathematical modeling) of the membrane separations (parameter and effect size estimation); (v) to optimize the objective function (the mathematical model) extracted from the analysis of the experimental design; and (vi) to develop an effective membrane cleaning method.

## 2. Materials and Methods

### 2.1. Beers

Canned Soproni Klasszikus pale lager beers (0.5 L) (HEINEKEN Hungária, Sopron, Hungary) with a 4.5% (*v*/*v*) ethanol content were used during the beer dealcoholization via reverse osmosis. The ingredients of this beer were water, malted barley, maize grits, hops, and hop extract.

### 2.2. Beer Dealcoholization via Reverse Osmosis

The BDA experiments were carried out with bench-scale “HF-528/08” crossflow reverse osmosis (CFRO) equipment (Hidrofilt, Nagykanizsa, Hungary) (Figure 2).

An RO99 flat sheet polyester membrane (Alfa Laval, Lund, Sweden) with a 0.05 m^2^ active surface was used for the dealcoholization processes. Dealcoholization experiments were performed according to the experimental design (Table 3) discussed in Section 2.9. The two factors were the transmembrane pressure (TMP) and the retentate flow rate (Q). Generally, in the case of RO processes, the retentate flow rate (Q) is lower than the feed flow rate by the permeate flow rate (flow drop) [23]. In this study, the permeate flow rates were less than 0.4% of the feed flow rates, and thus, the flow drops were negligible.

Before each dealcoholization experiment, in order to avoid foaming during the dealcoholization process, 5 L of beer (feed volume) was decarbonated by stirring for 30 min with an LR40 stirrer (MLW, Berlin, German Democratic Republic) with a marine propeller impeller with two blades at the lowest rpm to prevent vortex formation. After that, the water flux was measured at the given temperature and transmembrane pressure. Following the water flux measurement, in order to avoid the dilution of beer with water, the water from the CFRO equipment was drained with the valve at the bottom (Figure 2). Then, the residual water was carefully run off with beer.

The dealcoholization experiments were performed at a temperature of 15 ± 1 °C. During the filtrations, the pressures at both ends of the membrane module were measured. At the beginning of the filtrations, the first collected permeate samples (10 mL) were ignored to eliminate the dilution of permeate with water. For the rest of the time, the permeate samples were continuously collected at a constant volume (10 mL). Whenever the fluxes declined steadily and the required volumes of permeate samples were collected, the dealcoholization processes were finished at the same concentration factor. It should be noted that the properties of the beer samples did not change significantly because the concentration factor of the membrane separations was only 1.06. The durations of the dealcoholization processes were different because the concentration factors were the same.

### 2.3. Membrane Cleaning

The development process of a membrane cleaning method is detailed below. Based on the literature and from suggestions from the membrane manufacturers, an initial cleaning procedure was created. After the membrane separation process, this cleaning procedure was tested and modified. After the cleaning, the pure water flux was measured, and thus, the membrane cleaning efficiency could be calculated. If it was necessary, the types of chemicals, the concentration of the cleaning solutions, the temperature of the cleaning solutions, and the cleaning times were modified.

After each dealcoholization experiment, the membrane was cleaned thoroughly with deionized water for 10 min at a temperature of 25 °C, and then using 0.2% (*w*/*w*) sodium hydroxide for 60 min at a temperature of 25 °C. After cleaning with alkali, the membrane was rinsed again with deionized water for 10 min at a temperature of 25 °C. In all cases, the transmembrane pressure (TMP) and retentate flow rate (Q) were maintained at 6 bar and 240 L/h, respectively. Sodium hydroxide was purchased from Reanal, Budapest, Hungary. After each membrane cleaning, the water flux was measured at the given temperature and transmembrane pressure (TMP). The above-mentioned membrane cleaning procedure was developed based on the cleaning recommended by the membrane manufacturer (https://www.alfalaval.com/globalassets/documents/products/separation/membranes/flat-sheet-membranes/nf-and-ro-flat-sheet-membranes-200000076-3-en-gb.pdf?_ga=2.148687103.916594897.1626168162-1096977783.1626168162 accessed on 11 November 2022).

### 2.4. Analytical Parameters

The alcohol, real extract, and apparent extract contents of the beer and permeate samples were measured with Alcolyzer Plus (Anton-Paar, Graz, Austria). The bitterness levels (concentrations of iso-alpha acids in ppm) of the beer and the permeate samples were determined using the isooctane method [24]. The hydrogen chloride and isooctane for the bitterness measurements were purchased from Reanal, Hungary. The color of the beer and permeate samples were determined using the spectrophotometric method [25]. Absorbances were measured with a DR 6000 spectrophotometer (Hach, Ames, IA, USA), and a Heraeus Megafuge 16R Centrifuge (Thermo Fisher Scientific, Waltham, MA, USA) was used for the separation of the samples for the bitterness measurements. The pH values of the beer and permeate samples were determined with a 1100 H pH meter (VWR, Radnor, PA, USA). The turbidity values of the beer and the permeate samples were measured at a temperature of 20 °C (permanent haze) with a 2100P Turbidimeter (Hach, USA) in NTU, and converted to EBC [26]. The dynamic viscosity values of the beer and permeate samples were measured with a Physica MCR 51 Rheometer (Anton-Paar Hungary, Budapest, Hungary) with a DG27 double gap concentric cylinder measurement system. The data were acquired and analyzed using Rheoplus/32 software (Anton-Paar, Graz, Austria). The flow curves of the samples were measured by increasing the shear rate from 500 to 1000 1/s at temperatures of 0, 5, 10, 15, 20, and 25 °C. The dynamic viscosity values of the samples were calculated based on the Herschel–Bulkley model [27] fitted to the measured data of the flow curve (with shear stress as a function of the shear rate).

### 2.5. Separation Characteristics Parameters

Retentions of different components were calculated with Equation (1) [28]:(1)$$R_{i}=\left(1-\frac{C_{p_{i}}}{C_{b_{i}}}\right)\times 100$$
where *R_i_* is the retention (%) of the component *i*, $C_{p_{i}}$ (g/L) is the permeate concentration of the component *i*, and $C_{b_{i}}$ (g/L) is the bulk concentration of the component *i*.

### 2.6. Hydrodynamic Parameters

Water and permeate fluxes were determined with Equation (2) [29]:(2)$$J=\frac{V}{A_{m}\times t_{i}}$$
where J (L/m^2^ h) is the flux, V (L) is the permeate volume, $A_{m}$ (m^2^) is the membrane active surface area, and $t_{i}$ (h) is the time interval.

The permeate fluxes (mass-based) were determined with Equation (3) [11]:(3)$$J_{m}=\frac{m}{A_{m}\times t_{i}}$$
where $J_{m}$ (g/m^2^ h) is the mass flux, and m (g) is the permeate mass.

Ethanol fluxes were determined with Equation (4) (based on [30]):(4)$$J_{EtOH}=\frac{m_{EtOH}}{A_{m}\times t_{i}}=\frac{J\times c_{EtOH}\times \rho_{EtOH}}{100}$$
where $J_{EtOH}$ (g/m^2^ h) is the ethanol flux, $m_{EtOH}$ (g) is the mass of the ethanol in the permeate, $c_{EtOH}$ (% (*w*/*w*)) is the ethanol content in the permeate, and $\rho_{EtOH}$ (g/L) is the ethanol density at the given temperature.

To describe the flux during the early stage of the BDA process, a mathematical model (Equation (5)) was developed:(5)$$J_{t}=K\times t+J_{0}$$
where $J_{t}$ (L/m^2^ h) is the flux at any time (BDA permeate), $J_{0}$ (L/m^2^ h) is the initial flux (BDA permeate), K (1/h) is the flux decline coefficient (BDA permeate), and t (h) is the time.

Transmembrane pressures were determined with Equation (6) [31]:(6)$$TMP=\frac{p_{1}+p_{2}}{2}-p_{0}$$
where TMP (bar) is the transmembrane pressure, $p_{1}$ (bar) is the inlet pressure, $p_{2}$ (bar) is the outlet pressure, and $p_{0}$ (bar) is the pressure of the permeate.

Then, the intrinsic resistances of the clean membranes before the separations were determined with Equation (7) [31]:(7)$$J_{w\;0}=\frac{TMP}{\mu_{w}\times R_{m}}$$
where $J_{w\;0}$ (L/m^2^ h) is the water flux before separation, $\mu_{w}$ (Pas) is the dynamic viscosity of the water at the given temperature, and $R_{m}$ (1/m) is the intrinsic resistance of the clean membrane.

### 2.7. Evaluation of Cleaning Efficiency

Then, the intrinsic resistances of the membranes after the cleanings were determined with Equation (8) [32]:(8)$$J_{w\;w}=\frac{TMP}{\mu_{w}\times R_{n}}$$
where $J_{w\;w}$ (L/m^2^ h) is the water flux after the membrane cleaning, and $R_{n}$ (1/m) is the intrinsic resistance of the membrane after the membrane cleaning.

Flux recoveries were calculated using Equation (9) [32]:(9)$$FR=\frac{R_{m}}{R_{n}}\times 100$$
where FR is the flux recovery (%).

### 2.8. Linear Regression

Based on Equation (5) and on the time–flux data (with the exclusion of the first five unstable data points), the $J_{0}$ (for BDA via RO) and the K values of the seven individual filtrations (BDA process) were determined via regression, using IBM SPSS Statistics 25.0 software (IBM, Armonk, NY, USA). Significant differences in the parameter estimates, F values, and determination coefficients ($R^{2}$) of the models were evaluated. The normality of the residuals were accepted by the absolute values of their skewness and kurtosis, as they all were below 1 [33].

### 2.9. Modeling

The BDA via RO experiments were performed according to the $2^{p}$ full factorial experimental design [34] because the application of the experimental design minimizes the required number of experiments [35]. The aims of the application of the experimental design were as follows: (i) to formulate an objective function that describes the relationship between the factors and the response, and (ii) to determine the significant parameters and the effect sizes.

The general mathematical model for the $2^{3}$ full factorial experimental design (three factors, each at two levels) (Equation (10)) is as follows [34]:(10)$$Y=b_{0}+\sum_{i=1}^{3}b_{i}\times x_{i}+\sum_{i=1}^{3}\sum_{j=1,\;i\neq j}^{3}b_{ij}\times x_{i}\times x_{j}+b_{123}\times x_{1}\times x_{2}\times x_{3}$$
where Y is the response, $b_{0}$ is the constant, $b_{i}$ (i=1, 2, 3) are the regression coefficients of the main factor effects, $b_{ij}$ (i=1, 2, 3; j=1, 2, 3; i ≠ j), and $b_{123}$ are the regression coefficients of the interactions, and $x_{i}$ (i=1, 2, 3) are the coded factors.

The factors and levels of the $2^{p}$ full factorial experimental design are shown in Table 2.

The design matrix of the $2^{p}$ full factorial experimental design was generated in R 3.5.1 software (R Core Team, Vienna, Austria) using the RcmdrPlugin.DoE 0.12-3 package (Groemping and Fox), and it is shown in Table 3. The experiments were run in a random order to reduce the potential for biases.

The results of the experimental design were analyzed in various steps. First, the parameters of the objective functions were estimated (the non-significant parameters were eliminated), and model accuracies and determination coefficients were evaluated in R 3.5.1 software (R Core Team, Austria) using the RcmdrPlugin.DoE 0.12-3 package (Groemping and Fox). Secondly, after the standardization of the response values, the effect sizes of the significant parameters were calculated (linear regressions without constants), and the model accuracies and determination coefficients were evaluated in R 3.5.1 software (R Core Team, Austria) using the RcmdrPlugin.DoE 0.12-3 package (Groemping and Fox). Finally, the normalities of the residuals were checked using the Shapiro-Wilk normality test in RStudio 1.2.1335 software (RStudio Team, USA). According to the Shapiro-Wilk normality tests, the normalities of the residuals of the objective functions and the functions for the effect size determinations were accepted (p=0.72).

It was essential to find the global maximum of the objective function because the higher initial ethanol flux ($J_{EtOH\;0}$) is better from a technological point of view. The global optimization method ‘Grid Search’ [36] was used for this purpose. Aspects and comments on the Grid Search optimization method applied for the response objective function are shown in Table 4.

Based on the literature [37], the Grid Search algorithm was implemented in Scilab 6.1.0 software (ESI Group, Rungis, France). Furthermore, a 2D response plot of the effect of the significant parameter for the response was plotted in Scilab 6.1.0 software (ESI Group, France).

### 2.10. Assumptions

No unpredictable factors affected the course of the experiments. The equipment was functioning well, and no technical/equipment problems occurred. The samples were homogeneous (except for the volatile compound concentrations of the canned beers), and there were no sampling problems.

## 3. Results and Discussion

The following are the subsections in the Results and Discussion:**Analytical parameters** of beer and permeate samples,**Separation characteristic parameters** of BDA processes,**Hydrodynamic parameters** of BDA processes,Results of the **linear regression** of BDA processes,Results of **modeling,**Results of **membrane cleaning efficiency,**Research **limitations.**

### 3.1. Analytical Parameters

The measured analytical parameters of the beer (feed) (Section 2.1) are shown in Table 5.

Because of the high apparent attenuation (80%) of the lager yeast used, the final apparent extract was low. Generally, a lower final extract content can lead to lower fouling resistance and a lower osmotic pressure of the feed.

The bitterness of beer comes from a group of substances that are extracted components of hops during wort boiling [38]. The bitterness of the beer was not so high because the wort had probably been hopped moderately. About 20% of the phenolic compounds present in beer are derived from hops [39], and polyphenols are membrane foulants [40].

The color of the beer was pale, and the color of the beer is mostly attributed to melanoidins, a product of the final phase of the Maillard reaction [41]. Melanoidins have foam-stabilizing properties [42], and foaming can cause problems during membrane separation processes [43]. The color values of the permeate samples were 0.00 EBC. This means that the RO99 membrane (Section 2.2) completely rejected the color compounds of the beer.

The pH value of the beer was in the pH interval (4.2–4.4) of lager beers at the end of the fermentation [44]. The pH levels of the permeate samples (3.80 ± 0.01–4.07 ± 0.01) were slightly lower than the pH level of the beer. This may be because the acids of the beer passed through the RO99 membrane (Section 2.2).

According to the EBC standard (https://emin.com.mm/hannahi847492-02-hanna-hi847492-02-haze-meter-for-beer-quality-analysis-myanmar-30462/pr.html accessed on 11 November 2022), the beer was brilliant (0.0–0.5 EBC). Generally, if a beer is brilliant in terms of haziness, it leads to a lower fouling resistance. The turbidity values of the permeate samples were low (0.2–0.3 EBC), because the RO99 membrane (Section 2.2) rejected most of the haze-active compounds of the beer.

The dynamic viscosity values of the beer and permeate samples at the separation temperature are shown in Table 6.

The dynamic viscosity values were slightly high, and the reasons for this phenomenon are discussed below. The rotary viscometer was chosen because it provided a rapid measurement of the flow curve of the sample tested, with high reproducibility. The shear rate used in the test was rather high (when compared to the shear rate occurring in a falling ball or capillary viscometer), and therefore, the shear stress values were also higher. However, all of the samples showed Newtonian behavior (a linear flow curve). Furthermore, at lower temperatures, the dynamic viscosity values of the beer samples and permeate samples (ethanol–water mixture) were higher [45,46]. Therefore, the measured viscosity values (~5.9 mPas for the beer samples and ~5.2 mPas for the permeate samples) were appropriate values and were in the proper range (10^−3^ Pas).

The ethanol content values of the beer and permeate samples at 20 °C are shown in Table 7.

The alcohol content values of the permeate samples were low. Thus, the optimization of the operating parameters and the proper membrane area were required for the short dealcoholization process time. The alcohol content values of the permeate samples were similar to or lower than the values that can be found in the literature [11,14]. These differences are due to different operating parameters, membranes, and beer samples.

### 3.2. Separation Characteristics Parameters

The retention values of the real extract were ~99%, and the retention values of the Iso-alpha acids (bitterness) were 100% because of the application of the RO99 membrane (Section 2.2). The retention values of the Iso-alpha acids were lower in the literature than the values found in this study, because a membrane with a lower molecular weight cut-off (MWCO) was used in this study [12]. Furthermore, the organoleptic properties of the dealcoholized beer can be predicted well because, besides the measured analytical parameters of the permeate samples (color, pH value, turbidity, and dynamic viscosity), the calculated retention of the different components (real extract, Iso-alpha acids) significantly determined the sensory characteristics of this type of product.

### 3.3. Hydrodynamic Parameters

Figure 3 shows the hydrodynamic parameters of the separations.

The flux values were very low, especially the initial ethanol flux values. In order to obtain an adequate amount of permeate and separated ethanol, a large membrane area is required. However, compared to the flux values in other studies, the flux values were in the proper range [11,12,13,14]. The differences are due to different operating parameters, membranes, and beer samples.

### 3.4. Linear Regression

According to Student’s *t*-test, the parameter estimates were all highly significant in five cases (p<0.001). Similarly, the F values and $R^{2}$ values (F>27.9; $df_{1}=1$;  $24<df_{2}<26;$ p<0.001; $R^{2}>0.5$; p<0.001) of the models were also highly significant. For the settings ‘Standard order number 1’ and ‘Standard order number 5’, the parameter estimates were all significant, though less highly (p<0.05; p<0.01, respectively). In these two cases, we obtained some less but still significant F values ($F\left(1;23\right)>4.4$; p<0.05; $F\left(1;24\right)>14.7$; p<0.01, respectively) and $R^{2}$ values ($R^{2}$ >0.2; p<0.05;  $R^{2}>0.4;\;p<0.01$) as well.

### 3.5. Modeling

The parameter estimates of the significant parameters and the effect size estimate of the significant parameter of the objective function are shown in Table 8.

The TMP had a significant effect on the $J_{EtOH\;0}$. As can be seen in the literature [11], the effects of the TMP on the ethanol retention and permeate flux values were significant. Thus, it is clear as to why the TMP had a significant effect on the $J_{EtOH\;0}$. The Q had no significant effect on the $J_{EtOH\;0}$. As can be seen in the literature [11], the effect of Q on the ethanol retention and permeate flux values were close to negligible, with a wider Q range (Q: 120, 270, 420 L/h) than the applied Q range (Q: 120, 240 L/h) in this study. Thus, it is clear why the Q had no significant effect on the $J_{EtOH\;0}$. Furthermore, there was no significant interaction between the factors. From the final model, we omitted the Q and the interaction terms, and the significant coefficient of the TMP is represented in Table 8. The model accuracy and determination coefficients of the objective function were also significant ($F\left(1;5\right)=143.1;\;p<0.001$; multiple $R^{2}=0.97$; adjusted $R^{2}=0.96$). The objective function (Equation (11)) that exactly includes the parameters that are determined as significant in Table 8 is as follows:(11)$$J_{EtOH\;0}=80.871+41.094\times x_{TMP}$$

The linear model, which includes merely one factor (TMP), is quite simple and accurate at the same time.

A positive sign of the effect size indicates an interactive effect of the factors, while a negative sign of the effect size indicates an antagonistic effect of the factors. Thus, the TMP had an interactive effect on the $J_{EtOH\;0}$. The possible reasons for these phenomena are discussed below. Firstly, the difference of the TMP and the osmotic pressure difference is the driving force of RO. Therefore, a higher TMP led to a higher total initial flux (Section 3.3). Secondly, a higher TMP presses more foulants onto the membrane surface, forming a thicker fouling layer. The ethanol molecules could have been captured into the fouling layer. Therefore, a higher TMP led to higher ethanol retention (resulting in a lower ethanol concentration in permeate) (Section 3.1). In summary, it can be said that the higher TMP led to a higher total initial flux and a higher alcohol retention, but the effect of the TMP on the total initial flux was higher than the effect of the TMP on the ethanol retention. Thus, a higher TMP led to a higher $J_{EtOH\;0}$. The facts about the effect sizes of the TMP on the total initial flux and ethanol retention that are mentioned in this paragraph are not obvious.

The model accuracy and determination coefficients of the effect size estimation were significant ($F\left(1;6\right)=171.7;\;p<0.001$; multiple $R^{2}=0.97$; adjusted $R^{2}=0.96$).

Figure 4 shows the 2D response plot of the effect of the significant parameter ($x_{TMP}$) for the $J_{EtOH\;0}$.

The optimal value of the factor amounted to TMP = 30 bar, considering the ethanol flux. The predicted $J_{EtOH\;0}$ under the above condition was 121.965 g/m^2^ h. Therefore, the highest $J_{EtOH\;0}$ could be achieved with the highest TMP.

### 3.6. Membrane Cleaning Efficiency

The proposed cleaning method (Section 2.3) can be considered as being efficient because the average of the flux recoveries was 109%.

### 3.7. Limitations

During the BDA via RO experiments, the alcohol content limit (0.5% (*v*/*v*)) of beer was not reached because the process times of the dealcoholization trials would have been too long (measurable in days), due to the extremely low and continuously decreasing ethanol fluxes. Thus, the dealcoholization processes were carried out until the beginning of the preconcentration values at the same concentration factor. However, valuable information could be gained about the process.

The volatile fraction of the samples and dealcoholized beer could not be measured objectively. We tried to measure the volatile compounds of the samples, and the concentrations of the volatile compounds of the initial canned beer samples showed a large variance and inhomogeneity. Thus, the changes in the volatile compounds could not be determined properly because the concentrations of volatile compounds in the initial canned beer samples from the same batch were different.

According to the literature [47], the specific energy consumption of RO processes can be determined using a formula including the retentate flow rate (one of the factors of modeling in this study), the difference in the system pressure, and the permeate flux (one of the investigated parameters and optimized responses in this study). In this study, differences in the system pressure could not be determined exactly. Firstly, the difference in the system pressure was extremely low because of the small size of the membrane module, and thus, it could not be measured. Secondly, the difference in the system pressure could not be calculated because of the flat sheet design of the membrane module. Fortunately, the specific energy consumption can be deduced from the retentate flow rate value, the estimated difference in the system pressure, and the permeate flux.

The main objective of the study was not the statistical evaluation or validation of the measured analytical parameters. Only small amounts of samples could be collected, and thus, only a few parallel analytical experiments were conducted; statistical analyses with the aim of validation were not performed in these cases.

## 4. Conclusions

All of the goals of the BDA via RO investigation mentioned in Introduction (Section 1) were completely achieved: (i) valuable information for membrane separations was gained from the determined analytical parameters of beer, and the ethanol content values of the permeate samples could be used for the physical modeling; (ii) the determined values of the hydrodynamic parameters of the membrane separations could be used for the physical modeling; (iii) the calculated ethanol flux values of the membrane separations could be used for the physical modeling and the experimental design; (iv) the experimental design was analyzed, and the parameters of the objective function and effect sizes were estimated; (v) the global maximum of the objective function was successfully found, and the results of the optimization could directly be applied in practice; and (vi) an effective membrane cleaning method was developed.

The most important findings of the investigation are summarized, and the conclusions are drawn below. According to the analysis of the experimental design, the TMP had a significant effect, while the Q had no significant effect on the $J_{EtOH\;0}$ with the given parameters. Furthermore, there was no significant interaction between the factors. This means that commercial breweries should only focus on the optimization of the TMP. BDA via RO can be performed with the lowest required Q (Q = 120 L/h in our experiments), which results in a lower energy consumption. A lower energy consumption is important because of environmental and economic issues. Furthermore, the TMP had an interactive effect on the $J_{EtOH\;0}$.

Based on the results of the optimization, the highest $J_{EtOH\;0}$ ($J_{EtOH\;0}=121.965$ g/m^2^ h in our experiments) could be achieved with the highest TMP (TMP = 30 bar in our experiments). Thus, commercial breweries should set the TMP at the highest possible level, considering the energy consumption.

Furthermore, a slightly new and efficient (the flux recovery average was 109%) membrane cleaning method was developed and applied to recover the initial intrinsic resistance.

Our ethanol transport model can help with the optimization of industrial dealcoholization processes. In the future, the expansion of this model to various beer types and the transport of important aromatic compounds can expand its usefulness.

## Figures and Tables

**Figure 1 membranes-13-00329-f001:**
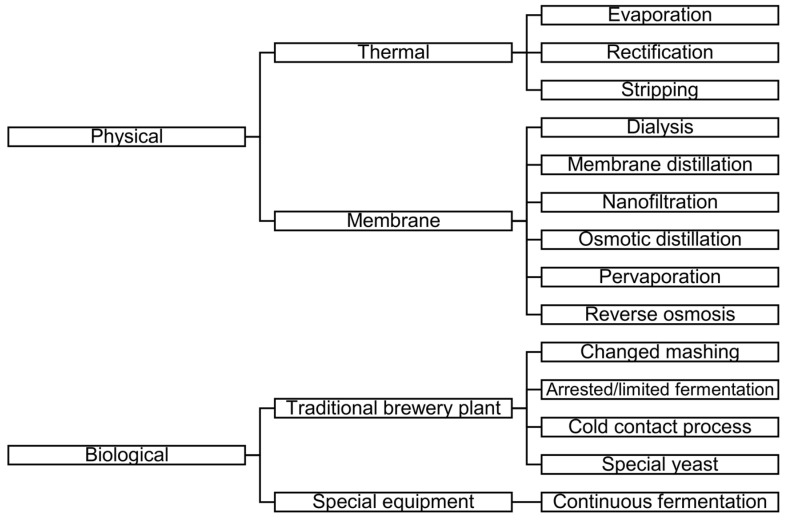
Schemes of low-alcohol beer (LAB) and alcohol-free beer (AFB) production methods (based on [3,5,6,7,8]).

**Figure 2 membranes-13-00329-f002:**
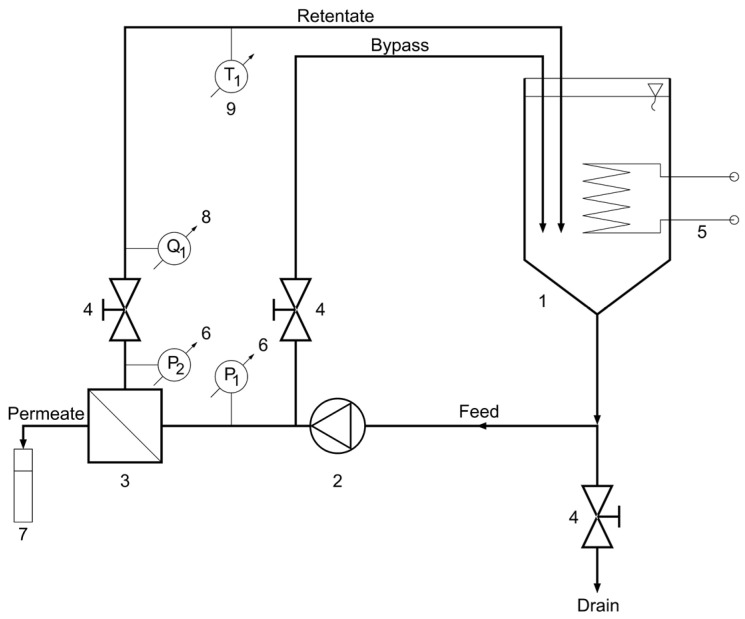
Schematic flow diagram of crossflow reverse osmosis (CFRO) equipment. 1—feed tank, 2—pump, 3—reverse osmosis membrane module, 4—valve, 5—heat exchanger (cooler/heater), 6—manometer, 7—measuring cylinder, 8—flowmeter, 9—thermometer.

**Figure 3 membranes-13-00329-f003:**
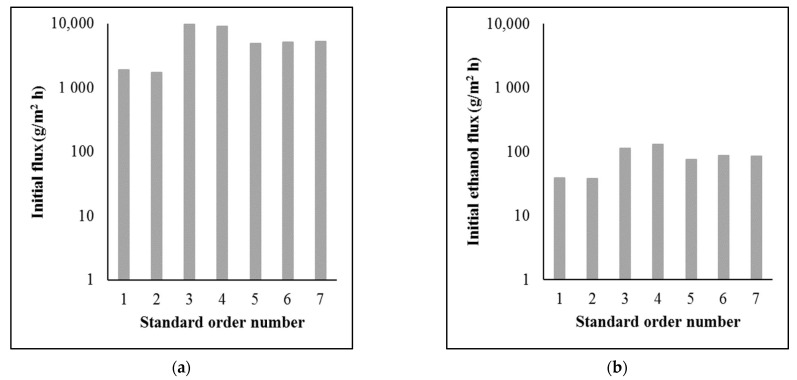
Hydrodynamic parameters of the separations: (**a**) initial flux (log); (**b**) initial ethanol flux (log).

**Figure 4 membranes-13-00329-f004:**
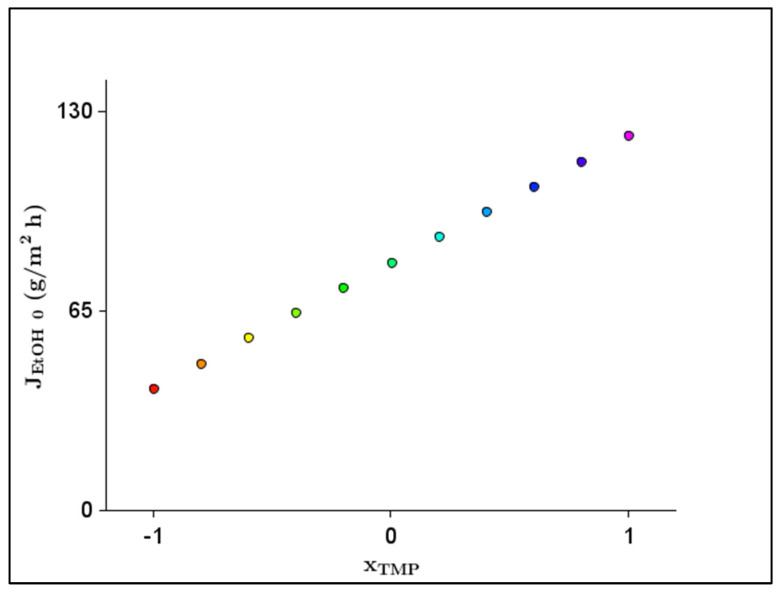
Two-dimensional response plot of the effect of significant parameter (transmembrane pressure–$x_{TMP}$) for initial ethanol flux ($J_{EtOH\;0}$).

**Table 1 membranes-13-00329-t001:** Comparison of installation cost, operating cost, and quality of final beer of low-alcohol beer (LAB) and alcohol-free beer (AFB) production methods (based on [10]).

Method	Installation Cost	Operating Cost	Beer Quality
Evaporation	**	*	*
Rectification	**	*	**
Stripping	**	**	**
Dialysis	***	*	*
Nanofiltration	**	**	***
Osmotic distillation	**	***	**
Pervaporation	*	**	***
Reverse osmosis	**	*	**
Changed mashing	*	***	*
Arrested/limited fermentation	*	***	*
Cold contact process	**	**	*
Special yeast	**	**	*
Continuous fermentation	*	**	*

*** More advantageous ** moderately advantageous, and * less advantageous.

**Table 2 membranes-13-00329-t002:** The factors and levels of the $2^{p}$ full factorial experimental design.

Factor	Abbreviation	Code	Unit	Factor Levels		
				Low (−1)	Central (0)	High (+1)
Transmembrane pressure	TMP	*x_TMP_*	bar	10	20	30
Retentate flow rate	Q	*x_Q_*	L/h	120	180	240

Initial ethanol flux ($J_{EtOH\;0}$) is the most important parameter of BDA via RO. Thus, $J_{EtOH\;0}$ was considered as the response of the $2^{p}$ full factorial experimental design of BDA via RO.

**Table 3 membranes-13-00329-t003:** The design matrix of the $2^{p}$ full factorial experimental design of BDA via RO experiments.

Standard Order Number	Actual Value			Coded Value
	TMP (bar)	Q (L/h)	x_TMP_	x_Q_
1	10	120	−1	−1
2	10	240	−1	+1
3	30	240	+1	+1
4	30	120	+1	−1
5 (C)	20	180	0	0
6 (C)	20	180	0	0
7 (C)	20	180	0	0

C = Center point.

**Table 4 membranes-13-00329-t004:** Aspects and comments on Grid Search optimization method applied for response objective function.

Method	Comments	Conclusion
Response method	The objective function is continuous.	Using Grid Search optimization for the response objective function can provide an optimal parameter set that can be directly applied in the membrane separation process.
	Analytical optimization of the objective function results in a parameter set that does not necessarily fit to the parameter settings available for membrane separation process.	
Grid Search optimization method	It is a numerical method with brute force (exhaustive) search (global optimization method on a grid).	
	It does not become stuck at a local optimum.	
	The set of optimization grid points can be adjusted to the resolution of the parameter ranges available for the membrane process.	

**Table 5 membranes-13-00329-t005:** Measured analytical parameters of the beer (feed).

Name of Parameter	Beer (Feed)
Alcohol content (*v*/*v*%)	4.34
Original real extract (*w*/*w*%)	10.28
Final real extract (*w*/*w*%)	3.63
Final apparent extract (*w*/*w*%)	2.04
Bitterness (IBU)	12
Color (EBC)	8.89
pH	4.23
Turbidity at 20 °C (EBC)	0.5
Dynamic viscosity at 20 °C (mPas)	5.48

**Table 6 membranes-13-00329-t006:** Dynamic viscosity values of beer and permeate samples at the separation temperature.

Sample		Dynamic Viscosity at 15 °C (mPas)
Beer (feed)		5.85 ± 0.03
Standard order number (permeate)	1	5.50 ± 0.03
	2	5.43 ± 0.01
	3	5.07 ± 0.04
	4	5.04 ± 0.03
	5	5.37 ± 0.03
	6	5.14 ± 0.02
	7	5.13 ± 0.02

**Table 7 membranes-13-00329-t007:** Ethanol content values of beer and permeate samples at 20 °C.

Sample		Ethanol Content at 20 °C (% (*v*/*v*))
Beer (feed)		4.34 ± 0.02
Standard order number (permeate)	1	2.56 ± 0.02
	2	2.75 ± 0.01
	3	1.45 ± 0.01
	4	1.82 ± 0.01
	5	1.92 ± 0.01
	6	2.10 ± 0.01
	7	2.07 ± 0.05

**Table 8 membranes-13-00329-t008:** Parameter estimates of the significant parameters, and effect size estimate of the significant parameter of the objective function.

Coefficient					Effect Size				
Term	Estimate	Standard Error	t	$Pr(>\left|t\right|)$	Parameter	Estimate	Standard Error	t	$Pr(>\left|t\right|)$
$b_{0}$	80.871	2.597	31.14	***	-	-	-	-	-
$b_{TMP}$	41.094	3.435	11.96	***	TMP	1.20389	0.09187	13.1	***

*** Significant at p<0.001.

## Data Availability

Not applicable.

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
