# Peer review of "Experimental Study and Modeling of Beer Dealcoholization via Reverse Osmosis"

_membranes, 2023, doi:10.3390/membranes13030329_

Round 1

Reviewer 1 Report

It would be interesting to know the volatile fraction of dealcoholized beer and how much does it cost per litre of beer this process.

Moreover, a critical point of comparison between reverse osmosis and other processes would be desirable to understand to what extent it can be a feasible solution for large and small breweries.

Author Response

Dear Reviewer1,

Thank you for your helpful advice and comments on the manuscript. We highly appreciate Your suggestions, which improved the quality of the work. Our responses are mentioned below.

English language and style of the manuscript has been edited by MDPI Author Services.

Comment 1

 It would be interesting to know the volatile fraction of dealcoholized beer

Answer 1

The volatile fraction of the samples and dealcoholized beer could not be measured objectively. We had tried to measure the volatile compounds of the samples and the concentrations of the volatile compounds of the initial canned beer samples showed a large variance and inhomogeneity. Thus, the changing of the volatile compounds can not be determined properly, because the concentrations of volatile compounds of initial canned beer samples from the same batch were different.

This extra information has been added to the Limitations section of the manuscript.

Comment 2

…how much does it cost per litre of beer this process.

Answer 2

You can see information about the cost of beer dealcoholized by reverse osmosis in Table 1 (installation cost, operating costs compared with different methods.), this Table has been added to the manuscript.

Comment 3

Moreover, a critical point of comparison between reverse osmosis and other processes would be desirable to understand to what extent it can be a feasible solution for large and small breweries.

Answer 3

You can see information about it, in the newly added Table 1.

As it can be seen in Table 1, BDA by RO is a slightly costly method, but the quality of the final beer is good compared to the other methods. Due to the higher costs, this method is a feasible solution for the larger breweries with capital.

Comment 4

The duration of the process is not clear.

Answer 4

It is written in the manuscript: “Whenever the fluxes declined steadily and the required volumes of permeate samples were collected, the dealcoholization processes were finished at the same concentration factor. It should be noted that the properties of the beer samples did not change significantly, because the concentration factors of the membrane separations were only 1.06.”

The time duration of the dealcoholization processes were different, because the concentration factors were the same. This extra information has been added to the manuscript.

Comment 5

The mathematical approach of the work is interesting, but the essential data for a company is missing, ie the process parameters. In the conclusions, for example, it is reported that the TMP must be set to the maximum.

I think it is necessary to give a value, or at least a range. Just as it is necessary when it comes to costs, to give some value, otherwise at the end of the work we are left without relevant information.

Answer 5

Values has been added to the Conclusion part of the manuscript.

Reviewer 2 Report

I have the following notes on the paper.
1. The introduction should be revised, citing data from other similar studies. In its current form, it gives rather some information without specifying why the research is being done.
2. Table 3 and its discussion should not be in Materials and Methods. The same applies to item 2.11. In my opinion, these items should be referred to the Results and Discussion.
3. The results and discussion section should also be revised. It is currently very fragmented and it is difficult to follow the logic of the results. Discussion of the results with other similar studies is missing or insufficient.

Author Response

Dear Reviewer2,

Thank you for your helpful advice and comments on the manuscript. We highly appreciate Your suggestions, which improved the quality of the work. Our responses are mentioned below.

English language and style of the manuscript has been edited by MDPI Author Services.

Comment 1

The introduction should be revised, citing data from other similar studies. In its current form, it gives rather some information without specifying why the research is being done.

Answer 1

It is important to note that scopes of several studies are BDA by RO [1–4]. However, none of the scopes of these studies are physical and mathematical modelling, and mathematical optimisation of BDA by RO. Thus, our study with its scope fills these research gaps, this was the reason why our research was done. Furthermore, Alfa Laval RO99 membrane has not been used for BDA according to the recent literature, the values and the ranges of the operating parameters in our study were different from the values and the ranges can be found in the recent literature and our beer sample (Soproni Klasszikus pale lager beer (HEINEKEN Hungária, Hungary)) was different from the beer samples that were dealcoholized in the recent literature [1–4].

These extra information and literature have been added to the manuscript.

Comment 2

Table 3 and its discussion should not be in Materials and Methods. The same applies to item 2.11. In my opinion, these items should be referred to the Results and Discussion.

Answer 2

Table 3 should not be referred to Results and Discussion, because this table confirms why the Grid Search optimization method was used. Section 2.11. was inserted into Results and Discussion Section.

Comment 3

The results and discussion section should also be revised. It is currently very fragmented and it is difficult to follow the logic of the results. Discussion of the results with other similar studies is missing or insufficient.

Answer 3

Extra information has been added to beginning of the Results and discussion section. Furthermore, more citations of different studies have been added to this section.

Literature for Reviewer2

  1. Catarino, M.; Mendes, A.; Madeira, L.M.; Ferreira, A. Alcohol Removal From Beer by Reverse Osmosis. Separation Science and Technology 2007, 42, 3011–3027, doi:10.1080/01496390701560223.
  2. Alcantara, B.M.; Marques, D.R.; Chinellato, M.M.; Marchi, L.B.; da Costa, S.C.; Monteiro, A.R.G. Assessment of Quality and Production Process of a Non-Alcoholic Stout Beer Using Reverse Osmosis. Journal of the Institute of Brewing 2016, 122, 714–718, doi:10.1002/jib.368.
  3. Ramsey, I.; Yang, Q.; Fisk, I.; Ayed, C.; Ford, R. Assessing the Sensory and Physicochemical Impact of Reverse Osmosis Membrane Technology to Dealcoholize Two Different Beer Styles. Food Chemistry: X 2021, 10, 100121, doi:10.1016/j.fochx.2021.100121.
  4. Jastrembska, K.; Jirankova, H.; Mikulasek, P. Dealcoholisation of Standard Solutions by Reverse Osmosis and Diafiltration. Desalin. Water Treat. 2017, 75, 357–362, doi:10.5004/dwt.2017.20544.

Round 2

Reviewer 2 Report

The authors had taken into account all previous comments.